# Towards Compute-Optimal Many-Shot In-Context Learning

**Shahriar Golchin**[1][*]    **Yanfei Chen**[2]    **Rujun Han**[2]    **Manan Gandhi**[2]    **Tianli Yu**[2]

**Swaroop Mishra**[3][†]    **Mihai Surdeanu**[1]    **Rishabh Agarwal**[3][‡]    **Chen-Yu Lee**[2]

**Tomas Pfister**[2]

[1]University of Arizona    [2]Google Cloud AI Research    [3]Google DeepMind

## Abstract

Long-context large language models (LLMs) are able to process inputs containing up to several million tokens. In the scope of in-context learning (ICL), this translates into using hundreds/thousands of demonstrations in the input prompt, enabling many-shot ICL. In practice, a fixed set of demonstrations is often selected at random in many-shot settings due to (1) high inference costs, (2) the benefits of caching and reusing computations, and (3) the similar performance offered by this strategy compared to others when scaled. In this work, we propose two straightforward strategies for demonstration selection in many-shot ICL that improve performance with minimal computational overhead. Our first method combines a small number of demonstrations, selected based on their similarity to each test sample, with a disproportionately larger set of random demonstrations that are cached. The second strategy improves the first by replacing random demonstrations with those selected using centroids derived from test sample representations via *k*-means clustering. Our experiments with Gemini Pro and Flash across several datasets indicate that our strategies consistently outperform random selection and surpass or match the most performant selection approach while supporting caching and reducing inference cost by up to an order of magnitude. We also show that adjusting the proportion of demonstrations selected based on *different criteria* can balance *performance* and *inference cost* in many-shot ICL.

## 1 Introduction

In-context learning (ICL) is a popular technique for adapting large language models (LLMs) to downstream tasks (Brown et al., 2020). With long-context LLMs (Anil et al., 2023; Georgiev et al., 2024; Fu et al., 2024; Ding et al., 2024) and the ability to cache and reuse computations (Pope et al., 2023), it has become practical to extremely increase the number of demonstrations in ICL, shifting from few-shot to many-shot settings to further enhance performance (Agarwal et al., 2024; Bertsch et al., 2024; Jiang et al., 2024, inter alia). In many-shot scenarios, random selection of demonstrations is often preferred (Agarwal et al., 2024; Bertsch et al., 2024), as it uses a fixed set of demonstrations without any selection criteria. This allows caching computations to control inference cost while achieving satisfactory performance. On the other hand, selection strategies based on specific criteria, such as similarity (Liu et al., 2021), often outperform random selection. However, such methods are impractical in many-shot scenarios, as they dynamically update the input prompt for each downstream test sample, which prevents caching and leads to substantial inference cost.

---

[*]Work done as a Student Researcher at Google Cloud AI Research. Correspondence to Shahriar Golchin <golchin@arizona.edu> and Chen-Yu Lee <chenyulee@google.com>.

[†]Now at Microsoft.

[‡]Now at Meta.

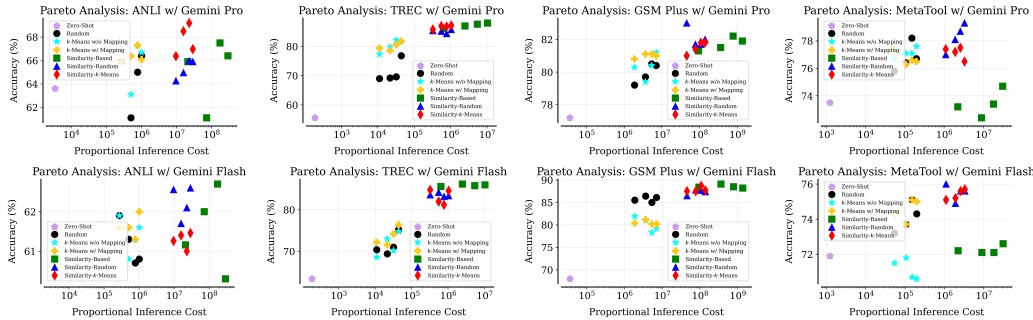

Figure 2: Pareto analysis of performance vs. inference cost across various datasets, comparing our hybrid selection strategies, i.e., similarity-random and similarity-*k*-means, with other methods. Our strategies balance the low inference cost of random or *k*-means-based selection with the performance gains of dynamic prompt-updating strategies, e.g., similarity-based selection, achieving results comparable to or better than the strongest selection approach.

We propose two novel strategies for demonstration selection in many-shot ICL. These strategies enable the dynamic customization of many-shot input prompt for each test sample while still remaining largely cacheable. Specifically, prompt customization is achieved by selecting a small number of demonstrations that are similar to each test sample, while the remaining demonstrations are chosen randomly and cached. This approach allows the cached random demonstrations to remain fixed across all test samples, significantly reducing computational overhead by requiring new computations *only* for the small number of similar demonstrations for each test sample. For example, in our selection strategy under a 100-shot setting, we select only 20 most similar demonstrations for each test sample and the other 80 random demonstrations remain cached. Figure 1 depicts this many-shot prompt format. This idea is motivated by an observation derived from analyzing results reported in previous studies (Agarwal et al., 2024; Bertsch et al., 2024): in many-shot settings, the influence of selection criteria (e.g., similarity) on performance diminishes as the number of demonstrations increases substantially, and *beyond a certain point*, their impact becomes nearly equivalent to that of random demonstrations. We further confirm this through our own experiments, presented in Subsection 4.2.

Building on the first strategy, our second strategy replaces the cached random demonstrations with a fixed, diverse set of demonstrations selected using *k*-means clustering. In particular, we compute centroids based on test sample representations, map these centroids to the representations of the available demonstrations, and select the most similar ones to cache for use during inference. Figure 3 illustrates this selection strategy.

The key contributions are as follows:

**(1)** We propose two novel strategies for demonstration selection in many-shot ICL that outperform or perform on par with the strongest selection approach while significantly reducing inference cost by making them largely cacheable.

**(2)** We show that LLMs benefit more from ICL when demonstrations are selected using multiple criteria, rather than relying on a single criterion such as similarity or diversity, and the proportion of demonstrations selected based on *multiple criteria* can balance *performance* and *inference cost*, as shown in Figure 2.

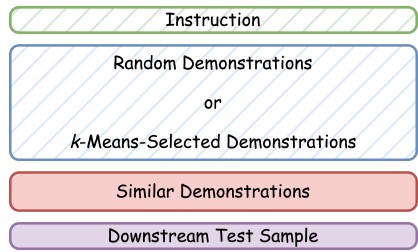

Figure 1: Our proposed many-shot ICL prompt format. Hatched blocks represent cached content, while solid blocks indicate non-cached content. When LLM is prompted, new computations are performed only for a small set of demonstrations selected based on similarity to the downstream test sample, while previous computations for a larger set of random or *k*-means-selected demonstrations are reused. Block sizes in the figure aim to reflect their actual proportions in the input prompt.

## 2 Approach

### 2.1 Motivation and Challenges

Many-shot ICL, due to using an extensive number of demonstrations, presents unique scenarios that make demonstration selection strategies used in few-shot settings impractical. Two key properties that characterize these scenarios are:

**(1) High Inference Cost:** Processing a large number of demonstrations in many-shot settings leads to substantial computational cost. Therefore, cacheable methods, such as random selection of demonstrations, are often preferred in such settings for practical use cases (Agarwal et al., 2024; Bertsch et al., 2024; Baek et al., 2024).

**(2) Diminishing Effectiveness of Selection Criteria:** Criteria that are effective for selecting demonstrations in few-shot settings lose their effectiveness on performance improvement as the number of demonstrations increases. For example, when demonstrations are chosen based on similarity, the sheer number of demonstrations causes this criterion to diminish gradually, and *beyond a certain point*, demonstrations effectively become equivalent to random selection (Agarwal et al., 2024; Bertsch et al., 2024). We verify this in Subsection 4.2.

Motivated by these two observations, we propose two new selection strategies tailored for many-shot scenarios. These approaches aim to address the challenges outlined above while combining the strengths of multiple selection criteria. Specifically, we construct a set of demonstrations that incorporate *multiple criteria,* such as similarity, diversity, and randomness, while maintaining the option for caching to keep inference costs manageable. The following subsections detail each selection strategy.

### 2.2 Hybrid Similarity-Random Selection

As noted, in many-shot settings, only a small number of demonstrations that are selected based on a specific criterion effectively improves performance. Given this insight, we select only a small set of demonstrations based on the similarity criterion to include in the input prompt. Specifically, we use $s$ most similar demonstrations to each downstream test sample.

To form the many-shot prompt, the remaining demonstrations are selected randomly. These random demonstrations are nearly as effective as similar ones in many-shot scenarios due to the diminishing effectiveness of selection criterion, with the added benefit of being fixed and therefore cacheable.

In this setup, as the number of random demonstrations is significantly larger than the number of similar ones, the majority of the input prompt remains fixed and cached, reducing the computational cost of inference in many-shot settings while tailoring the input prompt to each downstream test sample. In particular, an $n$-shot prompt is formed with $s$ similar demonstrations and $r$ random demonstrations, where $r \gg s$, and $n = s + r$. Figure 1 illustrates the approximate proportions of these components in the input prompt.[1]

### 2.3 Hybrid Similarity-$k$-Means Selection

As an alternative to our initial strategy, we use a $k$-means clustering approach to replace randomly selected demonstrations for the fixed and caching part of the input prompt. This method aims to enhance our first strategy by *encouraging diversity* in demonstration selection To select more effective demonstrations for test samples and avoid noisy demonstrations that can harm performance, we adapt the use of $k$-means clustering. Instead of directly applying it to the pool of available demonstrations, we first apply $k$-means clustering to the test samples and compute centroids using the elbow method (Thorndike, 1953). We then map these centroids to the representations of available demonstrations, selecting the ones most similar to the centroids. This ensures that the selected demonstrations are *semantically relevant* and *diverse* with respect to the test samples. Figure 3 depicts this mapping.

---

[1]See Appendix A for the actual prompts used for each dataset.

Finally, the number of demonstrations selected based on each centroid is determined by a predefined total, $c$. If the predefined total cannot be evenly split among centroids, the remaining demonstrations are randomly distributed across centroids until the total is met. Similar to our first strategy, for an $n$-shot prompt consisting of $s$ similar demonstrations and $k$ $k$-means-selected demonstrations, we maintain $k \gg s$, with $n = s + k$. Figure 1 displays these proportions as well.[2]

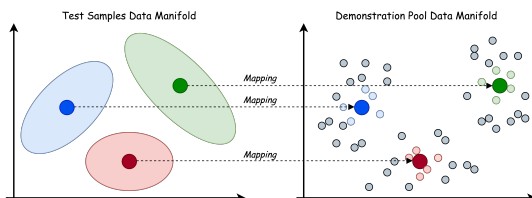

Figure 3: Illustration of selecting demonstrations using $k$-means clustering. First, centroids are computed based on the representations of test samples using the elbow method. These centroids are then mapped to the representations of available demonstrations. Lastly, demonstrations that are most similar to each centroid are selected based on a predefined total. In this example, the predefined total is 15 demonstrations, with three centroids identified. Therefore, each centroid is allocated five demonstrations.

## 3 Experimental Setup

**Data.** To ensure clarity across our evaluation settings, we outline the datasets used for each evaluation and refer to the corresponding subsections below.

Subsections 4.1, 4.2, and 4.3: To evaluate the effectiveness of our proposed selection methods, compare their inference costs with other approaches, and analyze the diminishing impact of selection criteria in many-shot ICL, we use tasks covering natural language inference, classification with high label complexity, reasoning, and tool use. Specifically, we utilize the ANLI (r3) (Nie et al., 2019), TREC (with 50 fine-grained labels) (Li & Roth, 2002; Hovy et al., 2001), GSM Plus (Li et al., 2024b), and MetaTool (Huang et al., 2023) datasets, respectively. For each dataset, we randomly select 1,000 test samples from the test set. If a test set has fewer than 1,000 samples, we use the entire set. From the train set, we sample up to 100,000 samples randomly; if fewer are available, we use the full train set. For datasets with only one split (e.g., only a test or train set), we allocate 1,000 random samples for testing and leverage the rest—up to 100,000 samples—as the train set. In all cases, the train set is viewed as the pool of available demonstrations.

Subsection 4.4: To assess our selection strategies in a low-data regime, we use selected tasks from the Big-Bench Hard (BBH) dataset (Suzgun et al., 2022), including Geometric Shapes, Logical Deduction (with seven objects), and Salient Translation Error Detection. For each task, we create a demonstration pool containing 100 random samples and a test set with 150 random samples.

**Models.** We employ Gemini Pro and Flash (Georgiev et al., 2024) as the base models in our experiments, accessed through Google API. We particularly use the `gemini-1.5-pro-002` and `gemini-2.0-flash-001` endpoints, respectively. To promote deterministic results, we set the temperature to zero across all experiments and maintain default values for all other decoding hyperparameters. For non-generative tasks, we cap the maximum completion length at 20 tokens; for generative tasks, we allow up to 8,000 tokens.

**Embeddings.** We leverage Gecko embeddings (Lee et al., 2024), accessed via Vertex AI on Google Cloud Platform, to select demonstrations based on similarity. Specifically, we utilize the `text-embedding-004` endpoint (Google Cloud, 2024) with a dimensionality of 768 and set the task type to `SEMANTIC SIMILARITY`. We compute similarity using cosine similarity and average embeddings.

**Demonstrations.** We vary the number of demonstrations from 0 to 200 in steps of 50, i.e., $n \in \{0, 50, 100, 150, 200\}$. Regardless of the value of $n$, a constant number of $s = 20$ similar demonstrations are always included in the input prompt across all settings when using our selection strategies. The remaining demonstrations are preselected either randomly or using $k$-means clustering, with $r, k \in \{0, 30, 80, 130, 180\}$, and cached for reuse during inference.

---

[2]In scenarios where test data is not available in advance, this method can be applied progressively as new test samples arrive.

As for the order of demonstrations, previous studies showed that the order in which demonstrations are presented in the input prompt can influence performance in *few-shot* ICL (Zhao et al., 2021; Lu et al., 2021; Zhang et al., 2022). However, this effect becomes negligible in *many-shot* settings (Bertsch et al., 2024). Therefore, for random selection baseline and our hybrid similarity-random selection strategy, both of which use randomly selected demonstrations, we present demonstrations in random order and do not study the impact of ordering, as it has little to no effect.

**Caching.** Our hybrid demonstration selection strategies do not depend on how caching is implemented, making them compatible with any caching approach. In this work, to cache content in many-shot settings, we choose to use the key-value caching method (Pope et al., 2023; Google AI, 2025), as it is widely adopted by the community and practitioners.

**Baselines.** In line with most previous studies on ICL (Dong et al., 2022; Agarwal et al., 2024; Bertsch et al., 2024), we utilize two commonly adopted demonstration selection strategies in many-shot ICL as our baselines: random and similarity-based selection.

In addition, we include two *k*-means-based baselines to evaluate the impact of mapping centroids from test data, as discussed in Subsection 2.2, and to assess the compound effect of incorporating similar demonstrations in our hybrid similarity-*k*-means strategy. The first baseline performs a centroid mapping from the test data to the pool of available demonstrations, then selects demonstrations most similar to those mapped centroids. This approach exactly follows the mapping process described in Subsection 2.2, except it excludes the similar demonstrations that are added for each downstream test sample. We refer to this baseline as *k-means with mapping*. The second baseline does not involve any centroid mapping from the test data. Instead, it directly identifies centroids within the pool of available demonstrations and selects demonstrations most similar to those centroids from that same pool. We call this baseline *k-means without mapping*. In both these baselines, the number of selected demonstrations from each cluster is evenly distributed based on the optimal number of clusters (i.e., centroids) and the total demonstration budget for each setting (e.g., 50-shot). If the demonstration budget cannot be evenly divided across clusters, the remaining are randomly assigned to clusters until the predefined total is reached.

We evaluate the performance of our proposed selection strategies against all *four baselines* and also compare the inference cost of each strategy in many-shot scenarios.

*k*-**Means Clustering.** In all our experiments, we use the default hyperparameters for *k*-means clustering provided by scikit-learn (Pedregosa et al., 2011). Additionally, we vary the number of clusters from 1 to 20 to determine the optimal number of clusters based on inertia and the elbow method. For our hybrid similarity-*k*-means approach and the *k*-means baseline with mapping, the optimal number of clusters is chosen using the test data. In contrast, for the *k*-means baseline without mapping, the optimal number of clusters is determined using the demonstration pool.

## 4 Results and Discussion

Figure 4 presents the many-shot ICL performance of Gemini Pro and Flash across four datasets, along with their inference cost comparisons. Figure 2 also displays the same results as Pareto plots. Below, we first compare our proposed demonstration selection strategies with other methods in terms of inference cost. Next, we examine how the impact of criteria used in demonstration selection strategies diminishes in many-shot ICL. We then evaluate the performance-cost efficiency of our methods against other approaches. Finally, we assess the effectiveness of our methods in low-data scenarios, where the pool of available demonstrations is limited.

### 4.1 Inference Cost Analysis

Due to the proprietary nature of our LLMs used in our experiments, we analyze proportional inference costs instead of absolute values for each method. Specifically, we calculate

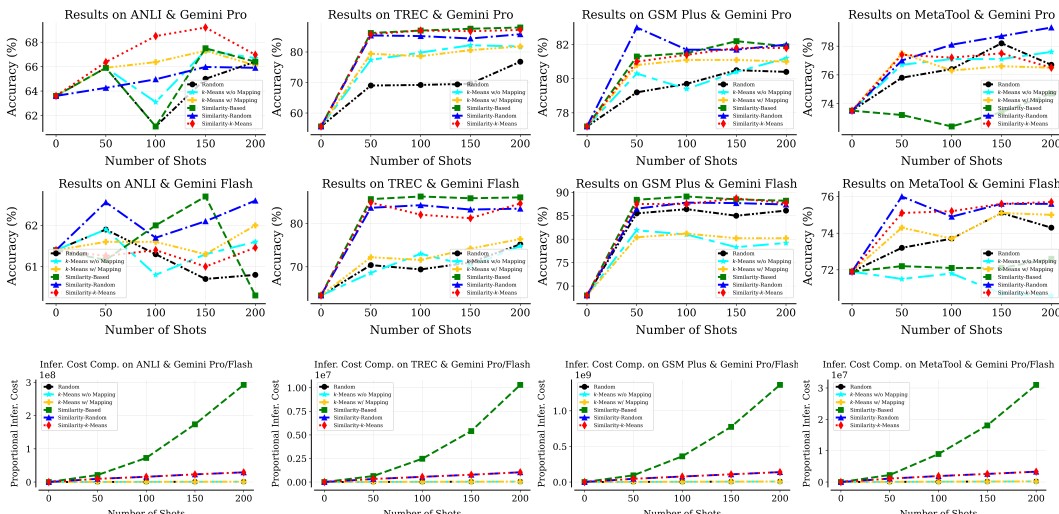

Figure 4: Results from Gemini Pro and Flash on various datasets in many-shot ICL settings. The first and second rows show performance plots for Gemini Pro and Flash, respectively, while the third row shows the inference cost comparisons for both models. We compare our two hybrid selection strategies in terms of both performance and inference cost against four baselines: (1) random selection, (2) $k$-means-based selection without mapping, (3) $k$-means-based selection with mapping, and (4) similarity-based selection.

proportional inference costs using the time complexity of the attention mechanism, as it is the predominant operation in Transformer-based models (Vaswani et al., 2017).

In the similarity-based selection strategy, where the input prompt is updated for each downstream test sample and thus cannot be cached, the proportional inference cost follows a complexity of $O(n^2)$, where $n$ represents the average number of tokens in the prompt.

When using cached content, however, inference cost is $O(ct + t^2)$, where $c$ is the average number of cached tokens, selected either randomly or via $k$-means clustering, and $t$ is the average number of tokens in the downstream test samples. This cost arises, as test tokens first attend to the cached content ($O(ct)$) and then to themselves ($O(t^2)$). However, in many-shot settings, where the average number of cached tokens is disproportionately larger than the average number of test tokens, i.e., $c \gg t$, inference cost is dominated by $O(ct)$.

Similarly, in our hybrid strategies, which dynamically add similar demonstrations to cached content, inference cost is $O(c(s + t) + (s + t)^2)$, where $s$ is the average number of tokens in the added similar demonstrations. Since each test sample is accompanied by a fixed number of similar demonstrations across *all* many-shot settings, and the average number of cached tokens disproportionately outnumber the average number of tokens from both similar demonstrations and test samples, i.e., $c \gg s + t$, inference cost simplifies to $O(c(s + t))$.

Therefore, our proposed hybrid selection strategies scale linearly with the average number of cached tokens, i.e., $O(c)$, similar to random or $k$-means-based selection methods with fully cached content. This linear scaling results in significantly lower inference cost compared to the similarity-based selection strategy, which is quadratic.

Note that, since we consider proportional inference costs rather than absolute values, model size does not affect the scaling—it acts as a constant multiplier. Also, all LLMs in our study share the same tokenizer, so the number of input tokens remains identical across different many-shot settings. This allows us to compare inference costs across different strategies and model sizes within the same figure for each dataset.

As shown in Figure 4, the similarity-based strategy, which updates the input prompt for each downstream test sample, leads to a quadratic increase in inference cost. This is especially

noticeable in tasks with long input contexts, such as in the GSM Plus dataset. In comparison, our hybrid methods maintain an inference cost similar to random or $k$-means-based selection, which is significantly lower than that of the similarity-based strategy.

## 4.2 Diminishing Impact of Selection Criterion

As discussed in Section 2, in many-shot ICL, the influence of selection criteria weakens as the number of demonstrations increases. This is visible in the performance plots shown in Figure 4. When similarity-based selection outperforms random selection, its benefit plateaus *beyond a certain point*. A similar pattern can be seen in both of the $k$-means-based baselines, where diversity is the key selection criterion.

For example, on the TREC dataset, the performance of the similarity-based selection strategy for both models remains almost unchanged beyond the 50-shot setting, whereas performance under random selection continues to improve, especially between 150 and 200 shots. In fact, similar demonstrations that contribute the most to improving performance are those selected initially. However, as more demonstrations are added, the similarity criterion decreases, reducing their contribution to performance improvement and resulting in almost no further gains. As a result, random selection begins to approximate the performance of similarity-based selection by including more demonstrations in the input prompt. This approximation is evident for the ANLI, GSM Plus, and MetaTool datasets, as indicated in Figure 4. For the TREC dataset, this approximation appears to require more than 200 shots, as shown by previous studies (Agarwal et al., 2024; Bertsch et al., 2024).

## 4.3 Performance-Cost Evaluation

Figure 2 presents Pareto plots for both Gemini Pro and Flash across different datasets, comparing our proposed selection strategies with other methods based on their proportional inference cost. Figure 4 further illustrates these metrics—performance and proportional inference cost—disaggregated by the number of demonstrations in the input prompt.

In a nutshell, random selection achieves the lowest overall performance and also incurs one of the lowest inference costs among all methods. The $k$-means-based selection strategy without centroid mapping performs slightly better at almost the same low inference cost, placing it second-lowest overall. When centroid mapping from test data is applied in the $k$-means strategy, performance improves further with almost no additional inference cost. This improvement shows that mapping centroids from test samples helps select more effective demonstrations and filter out noisy ones that can harm performance. In contrast, the similarity-based selection strategy generally achieves the highest performance among all baselines but results in significantly higher inference cost that grows quadratically.

Both of our selection strategies match the similarity-based approach in cases where it delivers the highest performance. In some datasets, such as ANLI and MetaTool, our strategies even outperform it. However, this level of performance in our methods is achieved with substantially lower inference costs—nearly two times lower in 50-shot settings and ten times lower in 200-shot settings. As discussed in Subsection 4.1, both of our methods have a computational complexity of $O(c)$, due to cached content, scaling linearly with the average number of cached tokens. In contrast, the similarity-based strategy has a complexity of $O(n^2)$, where $n$ is the average number of tokens in the input prompt. Overall, our methods effectively balance performance and inference cost, achieving comparable or better results compared to the similarity-based approach while keeping inference costs significantly lower.

When comparing our two proposed strategies, there is no single universally best performer across all datasets and settings. However, they have nearly identical inference costs, as both adopt the same approach to add similar demonstrations and cache the preselected ones.

More importantly, the performance plots in Figure 4 exhibit a compound effect of our hybrid selection strategies on both performance improvement and inference cost efficiency. For example, on the MetaTool dataset, the similarity-based strategy does not enhance performance compared to the zero-shot setting, resulting in almost no change in performance for both models. On the other hand, random selection of demonstrations enhances performance.

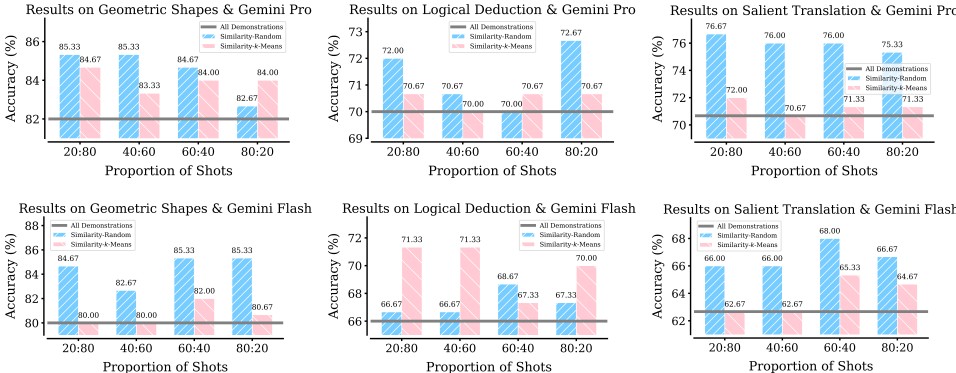

Figure 5: Results across various subsets of the BBH dataset in a 100-shot ICL setting under a low-data regime. While the total number of demonstrations is fixed at 100, the number of similar demonstrations increases from 20 to 80, moving from the leftmost to the rightmost bar pairs. Correspondingly, the number of demonstrations selected either randomly or via $k$-means decreases from 80 to 20. For example, in each plot, the leftmost pair of bars represents a setting with 20 similar demonstrations and 80 demonstrations selected either randomly (blue bar) or using $k$-means (red bar). Despite the limited pool of available demonstrations, our selection strategies outperform the common approach of using all available demonstrations in low-data scenarios.

When combined in our hybrid similarity-random selection strategy, performance exceeds that of random selection alone. Similarly, inference cost benefits from compound efficiency: random selection enables caching, while the similarity-based component adds only a small computational overhead of a few similar demonstrations.

A similar compound effect appears with our similarity-$k$-means strategy, as seen in Figure 4. For instance, on the TREC dataset, the performance of the $k$-means baseline with mapping plateaus after an initial increase. Likewise, the similarity-based strategy shows the same behavior but reaches a higher performance level. When the two methods are combined in our similarity-$k$-means strategy, the performance rises from the level of the $k$-means baseline with mapping to match that of the similarity-based strategy, which is the best-performing method. In the 50-shot setting, this improvement is more than 11% with Gemini Flash and over 6% with Gemini Pro, while the inference cost is cut in half.

As a result, the proportion of demonstrations selected based on each criterion can serve as a hyperparameter to balance performance and inference cost in many-shot ICL settings.

## 4.4 Low-Data Regime Evaluation

One key factor influencing the performance of a similarity-based selection strategy is the availability of a large and diverse demonstration pool to ensure that each downstream test sample has at least a few similar demonstrations to be included in the input prompt. To evaluate our approaches in scenarios with limited data, we test them under a condition where such a pool is unavailable. Maintaining ICL fixed at a 100-shot setting, we increase the number of similar demonstrations from 20 to 80 in increments of 20, while proportionally reducing the number of demonstrations selected either randomly or via $k$-means clustering.

This experiment answers two key questions commonly encountered in low-data regimes:

(1) How does the availability of similar demonstrations impact performance compared to using the full demonstration pool, the most common approach in low-data scenarios?

(2) In low-data scenarios where multiple runs allow for optimizing performance, how does adjusting the proportion of similar demonstrations influence performance?

Figure 5 demonstrates the results under the low-data regime. As shown, our selection strategies outperform the use of all available demonstrations, with the top-performing

method alternating between the two strategies across different datasets. This suggests that even when the demonstration pool is limited, our selection strategies are more effective. Furthermore, adjusting the proportion of similar demonstrations can lead to performance gains of about 3%–6% over using the full pool.

Additionally, as similar demonstrations are selected independently of those chosen randomly or via $k$-means, which is equivalent to selecting demonstrations with replacement, our findings complement those of Agarwal et al. (2024), who observed that repeating demonstrations in-context does not necessarily enhance performance and that distinct demonstrations are key to performance improvement. We argue that repetition can still be beneficial when the repeated demonstrations provide information relevant to the downstream test sample.

## 5  Related Work

**In-Context Learning.** ICL was first introduced by Brown et al. (2020) as a training-free method to adapt LLMs to downstream tasks for improved performance. However, the effectiveness of this technique heavily relies on the selection of demonstrations included in the input prompt (Bölücü et al., 2023; Wan et al., 2025). To address this limitation, various studies explored different criteria for selecting demonstrations, including similarity (Liu et al., 2021), diversity (An et al., 2023), or the ordering of demonstrations (Zhao et al., 2021; Lu et al., 2021; Zhang et al., 2022).

Another line of work aimed to uncover how ICL functions. Some studies analyzed its components to better understand its mechanism (Min et al., 2022; Yoo et al., 2022; Kossen et al., 2023; Lin & Lee, 2024). Others examined ICL through the perspective of gradient descent (von Oswald et al., 2023), while another view frames ICL as a method of compressing the training set into a single task-specific vector (Hendel et al., 2023). Moreover, Golchin et al. (2024) identified memorization as a key factor contributing to ICL.

More recent research on ICL discovered additional capabilities beyond enhancing performance. These capabilities include performing regression (Vacareanu et al., 2024), $k$-nearest neighbors (Agarwal et al., 2024; Dinh et al., 2022), jailbreaking (Anil et al., 2024), and LLMs as judges (Song et al., 2025; Golchin & Surdeanu, 2024; Li et al., 2024a, inter alia).

**Many-Shot In-Context Learning.** Along with the selection of demonstrations that influences ICL performance, the number of these demonstrations is another key factor that affects the results. With long-context LLMs, several studies attempted to mitigate this effect by significantly increasing the number of demonstrations to a few hundred (Zhang et al., 2023), thousands (Agarwal et al., 2024; Bertsch et al., 2024; Baek et al., 2024; Hao et al., 2022), or even by including the entire dataset within the input prompt as demonstrations (Agarwal et al., 2024; Baek et al., 2024; Wan et al., 2024).

Beyond LLMs, many-shot ICL was applied in other models and applications. For instance, it was used with multimodal foundation models to enhance downstream performance (Jiang et al., 2024) and was also employed in molecular inverse design (Moayedpour et al., 2024).

## 6  Conclusion

We proposed two straightforward yet effective demonstration selection strategies for many-shot in-context learning (ICL) that strike a balance between *performance* and *inference cost* by using *multiple criteria* to select demonstrations. To control inference cost, our methods cache a large set of demonstrations, preselected either randomly or via $k$-means clustering. Then, for each downstream test sample, a small number of similar demonstrations are dynamically included in the input prompt, further enhancing performance with only a modest computational overhead. This hybrid approach combines caching for efficiency with similarity-based selection for improved performance. Our experiments across various tasks indicated that our methods consistently exceed or match the best-performing selection strategy while reducing inference costs by up to an order of magnitude, making them both practical and scalable for large-scale many-shot ICL applications.

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

# A  Actual Input Prompts

Figures 6 to 19 display different parts of the actual input prompts used for each dataset, based on the prompt format shown in Figure 1. In these figures, even-numbered ones show the fixed, cached part of the input prompts, while odd-numbered figures illustrate the dynamic, uncached parts. Each cached and uncached pair combines to form the complete many-shot input prompt for a given dataset. For example, Figures 6 and 7 together form the complete input prompt used for the ANLI dataset.

The colors in these figures match the colors used in the prompt format in Figure 1. In particular, green highlights the instruction, blue represents demonstrations selected either randomly or using *k*-means clustering, red indicates similar demonstrations that are dynamically included in the input prompt for each downstream test sample, and purple shows the downstream test sample for which the model is prompted to make a prediction.

**Instruction:** Provide the most accurate label from the available labels that describes the relationship between the given sentence pair below.
Available Labels: ["Entailment", "Neutral", "Contradiction"]
– – –
**Premise:** Deshdrohi (English: Country Traitor) is a Bollywood comedy film. It was scripted and produced by Kamaal Rashid Khan who also appeared in the lead role with Manoj Tiwari, Hrishitaa Bhatt, Gracy Singh and Zulfi Syed. The movie has been listed as the worst Hindi movie ever by all the critics. The movie fared badly and people demanded double the amount paid as refund
**Hypothesis:** Country Traitor (Deshdrohi) is an inferior Hindi film
**Label:** Entailment
– – –
**Premise:** Oil prices, notoriously vulnerable to political events, spiked as high as $40 a barrel during the Gulf War in 1991.
**Hypothesis:** Oil prices will be affected by US elections.
**Label:** Neutral
– – –
[...] other {*Random Demonstrations*} or {*k-Means-Selected Demonstrations*}
– – –

Figure 6: An example of the fixed, cached part of the input prompt for the ANLI dataset. This prompt is followed by dynamic, uncached content, shown in Figure 7, to form the complete many-shot input prompt.

[...] previous {*Similar Demonstrations*}

– – –

**Premise:** The Cheap Flight Tony was trying to travel to his brother's wedding. He couldn't afford most tickets so he booked something cheap. He arrived for his flight bracing himself for the worst. The flight was long, dirty and uncomfortable. Still in the end at least he managed to travel safely to his destinati.

**Hypothesis:** This flight was an uncomfortable experience

**Label:** Entailment

– – –

**Premise:** I just let my thoughts run and I thought of the most surprising things . Marilla felt helplessly that all this should be sternly reproved , but she was hampered by the undeniable fact that some of the things Anne had said , especially about the minister 's sermons and Mr. Bell 's prayers , were what she herself had really thought deep down in her heart for years , but had never given expression to .

**Hypothesis:** Marilla and Anne are best friends.

**Label:** Neutral

– – –

**Premise:** I just let my thoughts run and I thought of the most surprising things . Marilla felt helplessly that all this should be sternly reproved , but she was hampered by the undeniable fact that some of the things Anne had said , especially about the minister 's sermons and Mr. Bell 's prayers , were what she herself had really thought deep down in her heart for years , but had never given expression to .

**Hypothesis:** Marilla did not say everything she felt.

**Label:**

Figure 7: An example of the dynamic, uncached part of the input prompt for the ANLI dataset. This portion of the prompt is updated for each downstream test sample.

**Instruction:** Provide the most accurate label from the available labels for the given text below.
Available Labels: ["Abbreviation", "Expression Abbreviated", "Animal", "Organ of Body",
"Color", "Invention Book and Other Creative Piece", "Currency Name", "Disease and Medicine",
"Event", "Food", "Musical Instrument", "Language", "Letter Like A-Z", "Other Entity", "Plant",
"Product", "Religion", "Sport", "Element and Substance", "Symbols and Sign", "Techniques and
Method", "Equivalent Term", "Vehicle", "Word with A Special Property", "Definition of
Something", "Description of Something", "Manner of An Action", "Reason", "Group or
Organization of Persons", "Individual", "Title of A Person", "Description of A Person", "City",
"Country", "Mountain", "Other Location", "State", "Postcode or Other Code", "Number of
Something", "Date", "Distance, Linear Measure", "Price", "Order Rank", "Other Number",
"Lasting Time of Something", "Percent Fraction", "Speed", "Temperature", "Size Area and
Volume", "Weight"]
– – –
**Text:** What TV show chronicled the lives of Katy Holstrum and Congressman Glen Morley ?
**Label:** Invention Book and Other Creative Piece
– – –
**Text:** How do fuel injectors work ?
**Label:** Manner of An Action
– – –
[...] other {*Random Demonstrations*} or {*k-Means-Selected Demonstrations*}
– – –

Figure 8: An example of the fixed, cached part of the input prompt for the TREC dataset.
This prompt is followed by dynamic, uncached content, shown in Figure 9, to form the
complete many-shot input prompt.

[...] previous {*Similar Demonstrations*}
– – –
**Text:** What Michelangelo sculpture is in Saint Peter 's Cathedral , Basilica , ?
**Label:** Invention Book and Other Creative Piece
– – –
**Text:** Who painted the Sistine Chapel ?
**Label:** Individual
– – –
**Text:** Who painted the ceiling of the Sistine Chapel ?
**Label:**

Figure 9: An example of the dynamic, uncached part of the input prompt for the TREC
dataset. This portion of the prompt is updated for each downstream test sample.

**Instruction:** Follow these instructions and solve the given question below: (1) solve the problem "step-by-step," (2) show all relevant calculations and reasoning, and (3) clearly state the final answer.

– – –

**Question:** Elly is arranging her books on some new bookshelves her parents gifted her. Each of the two central shelves can accommodate 10 books. The capacity of the lowest shelf is double that of a central shelf. The highest shelf can contain 5 books less than the lowest shelf. If she possesses 110 books, how many bookshelves will she require to store all of them?

**Solution:** The bottom shelf can hold 2 * 10 = <<2*10=20>>20 books.

The 2 middle shelves can hold a total of 2 * 10 = <<2*10=20>>20 books.

The top shelf can hold 20 – 5 = <<20-5=15>>15 books.

Each bookcase can hold a total of 20 books + 20 books + 15 books = <<20+20+15=55>>55 books.

To hold all of her books, she needs 110 / 55 = <<110/55=2>>2 bookcases.

#### 2

– – –

**Question:** Alicia's clothes have to be sent to the dry cleaners weekly. Her weekly drop-off includes 5 blouses, 2 pants, and 1 skirt. They charge her $5.00 per blouse, $6.00 per skirt, and $8.00 per pair of pants. She has a $3 coupon but finds it expired. How much does she spend on dry-cleaning in 5 weeks?

**Solution:** The blouses cost $5.00 each to dry clean and she has 5 so that's

5*5 = $<<5*5=25.00>>25.00

The one skirt she has costs $6.00 to dry clean

The pants cost $8.00 to dry clean and she has 2 pairs so that's 8*2 = $<<8*2=16.00>>16.00

In one week she spends 25+6+16 = $<<25+6+16=47.00>>47.00 on dry cleaning

Over 5 weeks, she will spend 5*47 = $<<5*47=235.00>>235.00 on dry cleaning

#### 235

– – –

[...] other {*Random Demonstrations*} or {*k-Means-Selected Demonstrations*}

– – –

Figure 10: An example of the fixed, cached part of the input prompt for the GSM Plus dataset. This prompt is followed by dynamic, uncached content, shown in Figure 11, to form the complete many-shot input prompt.

[...] previous {*Similar Demonstrations*}

– – –

**Question:** Raymond and Samantha share a familial bond as cousins. Raymond came into the world 6 years prior to Samantha. At the age of 23, Raymond became a father. Given that Samantha's current age is 31, can you determine how many years have passed since the birth of Raymond's son?

**Solution:** When Raymond's son was born Samantha was 23 - 6 = <<23-6=17>>17 years old. Thus it has been 31 - 17 = <<31-17=14>>14 years since Raymond's son was born.

#### 14

– – –

**Question:** Raymond and Samantha are cousins. Raymond was born 6 years and 3 months before Samantha. Raymond had a son at the age of 23. If Samantha is now 31, how many years ago was Raymond's son born?

**Solution:** 6 years and 3 months can be represented as 6 + 3/12 = 6 + 1/4 = 6.25 years. When Raymond's son was born Samantha was 23 - 6.25 = <<23-6.26=16.75>>16.75 years old. Thus it has been 31 - 16.75 = <<31-16.75=14.25>>14.25 years since Raymond's son was born.

#### 14.25

– – –

**Question:** Raymond and Samantha are cousins. Raymond was born 6 years before Samantha. Raymond had a son at the age of 23. Samantha's son is currently 7 years old, and Samantha gave birth to him at the age of 24. How many years ago was Raymond's son born?

**Solution:**

Figure 11: An example of the dynamic, uncached part of the input prompt for the GSM Plus dataset. This portion of the prompt is updated for each downstream test sample.

**Instruction:** You are a helpful and intelligent assistant.

Your task is to select the most appropriate tool to answer user's query.

Below is a list of available tools along with descriptions of when or where to use them.

You may only choose one tool from the list provided per query.

Available Tools: {

"timeport": "Begin an exciting journey through time, interact with unique characters, and learn history in this time-travel game!",

"airqualityforeast": "Planning something outdoors? Get the 2-day air quality forecast for any US zip code.",

"copilot": "Searches every dealer, analyzes & ranks every car for you so you can buy with confidence.",

"copywriter": "Send a URL and get sales copywriting suggestions for any page!",

"calculator": "A calculator app that executes a given formula and returns a result. This app can execute basic and advanced operations.",

**[...]**

}

– – –

**Query:** How can I save a copy of my conversation?

**Tool:** exportchat

– – –

**Query** I'm planning to buy a kitchen appliance from a reputable brand. Any recommendations?

**Tool:** ProductSearch

– – –

[...] other {*Random Demonstrations*} or {*k-Means-Selected Demonstrations*}

– – –

Figure 12: An example of the fixed, cached part of the input prompt for the MetaTool dataset. This prompt is followed by dynamic, uncached content, shown in Figure 13, to form the complete many-shot input prompt.

[...] previous {*Similar Demonstrations*}

– – –

**Query:** I'm not sure what career path to take. Can you help me find my dream job based on my interests and skills?

**Tool:** JobTool

– – –

**Query:** Hi, I'm curious about finding a job that aligns with my strengths and interests. Can you guide me in the right direction?

**Tool:** JobTool

– – –

**Query:** Can you suggest a specific method or tool that would help determine the type of job that would be the most suitable fit for my skills, interests, and qualifications?

**Tool:**

Figure 13: An example of the dynamic, uncached part of the input prompt for the MetaTool dataset. This portion of the prompt is updated for each downstream test sample.

**Instruction:** Given a full SVG path element containing multiple commands, your task is to determine the geometric shape that will be generated if one were to execute full path element. You may only generate the option letter corresponding to your answer, from options A to J.

– – –

**Input:** This SVG path element <path d="M 49.47,26.27 L 55.28,65.93 L 48.51,77.47 M 48.51,77.47 L 34.78,81.76 L 36.76,67.00 M 36.76,67.00 L 14.38,76.83 M 14.38,76.83 L 49.47,26.27"/> draws a:
(A) circle (B) heptagon (C) hexagon (D) kite (E) line (F) octagon (G) pentagon (H) rectangle
(I) sector (J) triangle
**Target:** C

– – –

**Input:** This SVG path element <path d="M 32.73,47.82 L 41.38,48.00 M 41.38,48.00 L 45.88,39.43 M 45.88,39.43 L 46.35,49.10 L 55.09,52.77 M 55.09,52.77 L 45.61,52.41 L 41.30,60.14 L 40.64,51.31 L 32.73,47.82"/> draws a:
(A) circle (B) heptagon (C) hexagon (D) kite (E) line (F) octagon (G) pentagon (H) rectangle
(I) sector (J) triangle
**Target:** F

– – –

[...] other {*Random Demonstrations*} or {*k-Means-Selected Demonstrations*}

– – –

Figure 14: An example of the fixed, cached part of the input prompt for the BBH dataset, Geometric Shapes subset. This prompt is followed by dynamic, uncached content, shown in Figure 15, to form the complete many-shot input prompt.

[...] previous {*Similar Demonstrations*}

– – –

**Input:** This SVG path element <path d="M 55.57,80.69 L 57.38,65.80 M 57.38,65.80 L 48.90,57.46 M 48.90,57.46 L 45.58,47.78 M 45.58,47.78 L 53.25,36.07 L 66.29,48.90 L 78.69,61.09 L 55.57,80.69"/> draws a:
(A) circle (B) heptagon (C) hexagon (D) kite (E) line (F) octagon (G) pentagon (H) rectangle
(I) sector (J) triangle
**Target:** B

– – –

**Input:** This SVG path element <path d="M 55.58,17.52 L 53.95,26.14 L 47.22,29.95 M 47.22,29.95 L 48.21,22.28 L 55.58,17.52"/> draws a:
(A) circle (B) heptagon (C) hexagon (D) kite (E) line (F) octagon (G) pentagon (H) rectangle
(I) sector (J) triangle
**Target:** D

– – –

**Input:** This SVG path element <path d="M 55.46,58.72 L 70.25,50.16 M 70.25,50.16 L 78.35,57.33 M 78.35,57.33 L 71.18,65.42 L 55.46,58.72"/> draws a:
(A) circle (B) heptagon (C) hexagon (D) kite (E) line (F) octagon (G) pentagon (H) rectangle
(I) sector (J) triangle
**Target:**

Figure 15: An example of the dynamic, uncached part of the input prompt for the BBH dataset, Geometric Shapes subset. This portion of the prompt is updated for each downstream test sample.

**Instruction:** Deduce the correct order of a sequence of objects based on the clues and information provided about their spacial relationships and placements.
You may only generate the option letter corresponding to your answer, from options A to G.
– – –
**Input:** The following paragraphs each describe a set of seven objects arranged in a fixed order. The statements are logically consistent within each paragraph. A fruit stand sells seven fruits: mangoes, watermelons, peaches, kiwis, oranges, cantaloupes, and plums. The watermelons are the cheapest. The peaches are more expensive than the mangoes. The cantaloupes are the second-most expensive. The oranges are more expensive than the cantaloupes. The peaches are less expensive than the plums. The kiwis are the third-cheapest.
(A) The mangoes are the third-most expensive
(B) The watermelons are the third-most expensive
(C) The peaches are the third-most expensive
(D) The kiwis are the third-most expensive
(E) The oranges are the third-most expensive
(F) The cantaloupes are the third-most expensive
(G) The plums are the third-most expensive
**Target:** G
– – –
**Input:** The following paragraphs each describe a set of seven objects arranged in a fixed order. The statements are logically consistent within each paragraph. On a shelf, there are seven books: a purple book, a green book, a white book, a gray book, a red book, a black book, and a brown book. The gray book is to the left of the purple book. The white book is to the right of the brown book. The black book is the third from the right. The purple book is to the left of the white book. The white book is the second from the right. The gray book is the third from the left. The brown book is to the right of the green book.
(A) The purple book is the third from the left
(B) The green book is the third from the left
(C) The white book is the third from the left
(D) The gray book is the third from the left
(E) The red book is the third from the left
(F) The black book is the third from the left
(G) The brown book is the third from the left
**Target:** D
– – –
[...] other {*Random Demonstrations*} or {*k-Means-Selected Demonstrations*}
– – –

Figure 16: An example of the fixed, cached part of the input prompt for the BBH dataset, Logical Deduction subset. This prompt is followed by dynamic, uncached content, shown in Figure 17, to form the complete many-shot input prompt.

[...] previous {*Similar Demonstrations*}

– – –

**Input:** The following paragraphs each describe a set of seven objects arranged in a fixed order. The statements are logically consistent within each paragraph. On a branch, there are seven birds: an owl, a crow, a falcon, a cardinal, a hummingbird, a quail, and a hawk. The falcon is to the left of the crow. The quail is to the right of the cardinal. The hummingbird is to the right of the quail. The falcon is the second from the right. The hummingbird is to the left of the hawk.
(A) The owl is the second from the right
(B) The crow is the second from the right
(C) The falcon is the second from the right
(D) The cardinal is the second from the right
(E) The hummingbird is the second from the right
(F) The quail is the second from the right
(G) The hawk is the second from the right
**Target:** C

– – –

**Input:** The following paragraphs each describe a set of seven objects arranged in a fixed order. The statements are logically consistent within each paragraph. On a branch, there are seven birds: an owl, a crow, a falcon, a cardinal, a hummingbird, a quail, and a hawk. The falcon is to the left of the crow. The quail is to the right of the cardinal. The hummingbird is to the right of the quail. The falcon is the second from the right. The hummingbird is to the left of the hawk.
(A) The owl is the second from the left
(B) The crow is the second from the left
(C) The falcon is the second from the left
(D) The cardinal is the second from the left
(E) The hummingbird is the second from the left
(F) The quail is the second from the left
(G) The hawk is the second from the left
**Target:** F

– – –

**Input:** The following paragraphs each describe a set of seven objects arranged in a fixed order. The statements are logically consistent within each paragraph. On a branch, there are seven birds: an owl, a crow, a falcon, a cardinal, a hummingbird, a quail, and a hawk. The falcon is to the left of the crow. The quail is to the right of the cardinal. The hummingbird is to the right of the quail. The falcon is the second from the right. The hummingbird is to the left of the hawk.
(A) The owl is the fourth from the left
(B) The crow is the fourth from the left
(C) The falcon is the fourth from the left
(D) The cardinal is the fourth from the left
(E) The hummingbird is the fourth from the left
(F) The quail is the fourth from the left
(G) The hawk is the fourth from the left
**Target:**

Figure 17: An example of the dynamic, uncached part of the input prompt for the BBH dataset, Logical Deduction subset. This portion of the prompt is updated for each downstream test sample.

**Instruction:** Given a source sentence written in German and its translation in English, your task is to determine the type of translation error that the translated sentence contains.
You may only generate the option letter corresponding to your answer, from options A to F.
– – –
**Input:** The following translations from German to English contain a particular error. That error will be one of the following types: Named Entities: An entity (names, places, locations, etc.) is changed to a different entity. Numerical Values: Numerical values (ordinals or cardinals), dates, and/or units are changed. Modifiers or Adjectives: The modifiers and adjectives pertaining to a noun are changed. Negation or Antonyms: Introduce or remove a negation or change comparatives to their antonyms. Facts: Trivial factual errors not pertaining to the above classes are introduced in the translations. Dropped Content: A significant clause in the translation is removed. Please identify that error.
Source: Ruth Lynn Deech, Baroness Deech DBE ist eine britische Juristin und Hochschullehrerin, die seit 2005 als Life Peeress Mitglied des House of Lords ist.
Translation: Ruth Lynn Deech, Baroness Deech DBE is a British lawyer and university lecturer who has been a life peer of the House of Lords since 2015.
The translation contains an error pertaining to
(A) Modifiers or Adjectives
(B) Numerical Values
(C) Negation or Antonyms
(D) Named Entities
(E) Dropped Content
(F) Facts
**Target:** B
– – –
**Input:** The following translations from German to English contain a particular error. That error will be one of the following types: Named Entities: An entity (names, places, locations, etc.) is changed to a different entity. Numerical Values: Numerical values (ordinals or cardinals), dates, and/or units are changed. Modifiers or Adjectives: The modifiers and adjectives pertaining to a noun are changed. Negation or Antonyms: Introduce or remove a negation or change comparatives to their antonyms. Facts: Trivial factual errors not pertaining to the above classes are introduced in the translations. Dropped Content: A significant clause in the translation is removed. Please identify that error.
Source: Štramberk ist eine Stadt in Tschechien.
Translation: it is a town in the Czech Republic.
The translation contains an error pertaining to
(A) Modifiers or Adjectives
(B) Numerical Values
(C) Negation or Antonyms
(D) Named Entities
(E) Dropped Content
(F) Facts
**Target:** D
– – –
[...] other {*Random Demonstrations*} or {*k-Means-Selected Demonstrations*}
– – –

Figure 18: An example of the fixed, cached part of the input prompt for the BBH dataset, Salient Translation subset. This prompt is followed by dynamic, uncached content, shown in Figure 19, to form the complete many-shot input prompt.

[...] previous {*Similar Demonstrations*}

– – –

**Input:** The following translations from German to English contain a particular error. That error will be one of the following types: Named Entities: An entity (names, places, locations, etc.) is changed to a different entity. Numerical Values: Numerical values (ordinals or cardinals), dates, and/or units are changed. Modifiers or Adjectives: The modifiers and adjectives pertaining to a noun are changed. Negation or Antonyms: Introduce or remove a negation or change comparatives to their antonyms. Facts: Trivial factual errors not pertaining to the above classes are introduced in the translations. Dropped Content: A significant clause in the translation is removed. Please identify that error.

Source: Ekkehard Drefke war ein deutscher Künstler.

Translation: Eduard Drefke was a German artist.

The translation contains an error pertaining to

(A) Modifiers or Adjectives

(B) Numerical Values

(C) Negation or Antonyms

(D) Named Entities

(E) Dropped Content

(F) Facts

**Target:** D

– – –

**Input:** The following translations from German to English contain a particular error. That error will be one of the following types: Named Entities: An entity (names, places, locations, etc.) is changed to a different entity. Numerical Values: Numerical values (ordinals or cardinals), dates, and/or units are changed. Modifiers or Adjectives: The modifiers and adjectives pertaining to a noun are changed. Negation or Antonyms: Introduce or remove a negation or change comparatives to their antonyms. Facts: Trivial factual errors not pertaining to the above classes are introduced in the translations. Dropped Content: A significant clause in the translation is removed. Please identify that error.

Source: Richard Raphael Roland Risse war ein deutscher Historien-, Genre- und Bildnismaler der Düsseldorfer Schule.

Translation: Risse was a German historical, genre and portrait painter of the Düsseldorf School.

The translation contains an error pertaining to

(A) Modifiers or Adjectives

(B) Numerical Values

(C) Negation or Antonyms

(D) Named Entities

(E) Dropped Content

(F) Facts

**Target:**

Figure 19: An example of the dynamic, uncached part of the input prompt for the BBH dataset, Salient Translation subset. This portion of the prompt is updated for each downstream test sample.

