# OpenReview forum: "Towards Compute-Optimal Many-Shot In-Context Learning"
_colmweb.org/COLM/2025/Conference — COLM 2025_

### Official Review · Reviewer_nzUp · 2025-05-03

**Rating:** 7
**Confidence:** 4
**Ethics Flag:** 1

**Summary:**

This paper incorporates caching into many-shot ICL, greatly shortening the inference cost. It is based on an observed phenomenon that example selection becomes less beneficial as the number of examples increases. Then, it fixes the majority of selected examples because they can be cached. The experiments tested on Gemini Pro and Flash show this method performs on par with or outperforms random and similarity-based baselines.

**Questions To Authors:**

- In Figure 5, why does hybrid similarity-random outperform hybrid similarity-k-means most of the time? Intuitively, involving k-means should be no worse than random selection, but your experiments give a reverse conclusion.
- How do you measure Inference cost in Figure 4? Since there are two models tested, why is there only one line for each method?

**Reasons To Accept:**

- This paper leverages an interesting fact that example selection becomes less important as the number goes up. The proposed performs as expected, reducing inference cost greatly.
- The whole method and its implementations are easy to understand.
- It conducts comprehensive experiments to demonstrate the effectiveness of proposed methods.

**Reasons To Reject:**

- The tested models are both from the Gemini family, but it doesn't show any results on other LLMs, especially the open-source models like llama.
- Some experiment results seem to contradict one of the motivations. In Line 84-87, it states that similarity-based selection effectively becomes equivalent to random selection. However, in Figure 4, the performance gap between these two methods can be as large as 10%+ accuracy (see TREC dataset). The statement in Line 230-232 becomes "performance for both models remains almost unchanged beyond the 50-shot setting".

---

> ### Author Response · Authors · 2025-06-02
> **Official Response by Authors (1/2)**
>
> Thank you for recognizing the effectiveness and straightforward nature of our proposed strategies, as well as the wide range of experiments used in our study. We appreciate your insightful feedback and the opportunity to respond to your concerns in detail. We address each of your points one by one in the following, in the order they are presented.
>
> > The tested models are both from the Gemini family, but it doesn't show any results on other LLMs, especially the open-source models like llama.
>
> Thank you for your input. We would like to highlight that **our choice of models and datasets was guided by the computational resources available to us, a strategy followed by all prominent prior works on many-shot ICL, such as Agarwal et al. (2024) and Bertsch et al. (2025).** For example, Agarwal et al. (2024) primarily used a single LLM (Gemini 1.5 Pro) in their experiments, while Bertsch et al. (2025) mostly focused on classification datasets. *Informed by these precedents, we did our best to carefully balance the number of models and datasets, ensuring diversity in both model size and task type. Specifically, we selected two LLMs of significantly different sizes, and our dataset choices span a range of tasks, from complex classification to tool use, reflecting practical applications valued by the community.*
>
> As a result, given our available resources, **we strongly believe that our study includes a sufficiently diverse set of models, datasets, and scenarios to effectively demonstrate the benefits of our proposed demonstration strategies in many-shot settings.**
>
> > Some experiment results seem to contradict one of the motivations. In Line 84-87, it states that similarity-based selection effectively becomes equivalent to random selection. However, in Figure 4, the performance gap between these two methods can be as large as 10%+ accuracy (see TREC dataset). The statement in Line 230-232 becomes "performance for both models remains almost unchanged beyond the 50-shot setting".
>
> Thank you for your input. We would like to clarify that our claim—that *“similarity-based selection effectively becomes equivalent to random selection”*—is **_always_** accompanied by the important qualifier **_“beyond a certain point.”_** This qualifier is explicitly included in the relevant parts of the paper, specifically in lines 45–49 and 84–87, to emphasize that **this approximation occurs only after a certain threshold.** Our results, shown in Figure 4, clearly support this claim, as do previous studies such as Agarwal et al. (2024) and Bertsch et al. (2025).
>
> **Regarding the TREC dataset,** it can be seen in Figure 4 that the performance of the random selection strategy begins to converge to the similarity-based strategy as the number of demonstrations increases—particularly from 150 to 200. However, **for this dataset, the point at which similarity-based selection starts to closely approximate random selection seems to occur beyond 200 demonstrations. This convergence point is not visible in our current results due to the experimental limit of 200 demonstrations. Nevertheless, this does not invalidate our claim,** and the TREC dataset still falls within the scope of our assertion. In fact, *as shown in the work of Bertsch et al. (2025), specifically in Figure 2b, further increasing the number of demonstrations leads to a close approximation between the two strategies for this dataset.* **Therefore, our claims in the paper are fully consistent with the reported results, and there is no contradiction between our assertion and the data.**
>
> ---
>
> ### **Questions:**
>
> > In Figure 5, why does hybrid similarity-random outperform hybrid similarity-k-means most of the time? Intuitively, involving k-means should be no worse than random selection, but your experiments give a reverse conclusion.
>
> Thank you for your question. We would like to emphasize that the effectiveness of one selection strategy compared to another depends on several factors, such as the downstream task, pretraining/fine-tuning data, training recipe, model size, and more. For example, as shown in Figure 4, the hybrid similarity-random method performs better than the hybrid similarity-$k$-means method on the ANLI dataset using Gemini Flash. However, when using Gemini Pro on the same dataset, the results are reversed. A similar but opposite trend can also be seen with the MetaTool dataset. Therefore, there is no strict rule to guarantee that one specific method will consistently perform better than the other.

---

> > ### Author Response · Authors · 2025-06-02
> > **Official Response by Authors (2/2)**
> >
> > > How do you measure Inference cost in Figure 4? Since there are two models tested, why is there only one line for each method?
> >
> > Thank you for your question. As we detailed in lines 216–220 of the paper, we compare inference costs proportionally, rather than using absolute values. In this case, model size does not affect this cost as it only acts as a multiplier. Further, both models we used in our study share the same tokenizer. Therefore, the number of input tokens in all our experiments for each specific $n$-shot ICL setting is identical across both models. This allows us to compare inference costs across different selection strategies and model sizes within the same figure for each dataset.
> >
> > ---
> >
> > In closing, we hope our responses have addressed your concerns effectively. We are always happy and open to further discussion if you still have any concerns or questions.
> >
> > ---
> >
> > **References:**
> >
> > [1] Many-Shot In-Context Learning (Agarwal et al., NeurIPS 2024)
> >
> > [2] In-Context Learning with Long-Context Models: An In-Depth Exploration (Bertsch et al., NAACL 2025)

---

> > > ### Comment · Reviewer_nzUp · 2025-06-05
> > >
> > > Thank you for the clarification. I think the clarification on the *reasons to reject* is fair enough, and I have raised my score. Nevertheless, I should say this work will become more influential if you experiment on Llama models (just a suggestion).
> > >
> > > Regarding Figure 5, I still expect to see conclusions about when **Hybrid Similarity-Random** is better and when "Hybrid Similarity-k-Means" is better. Instead of merely demonstrating "at least one of your methods outperforms the **All Demonstrations** setting".

---

> > > > ### Author Response · Authors · 2025-06-06
> > > > **Official Response by Authors**
> > > >
> > > > First of all, we would like to thank you for your engagement during the discussion period. We are glad that our responses have addressed your concerns.
> > > >
> > > > We would now like to address your recent suggestion and earlier question as follows:
> > > >
> > > > > Nevertheless, I should say this work will become more influential if you experiment on Llama models (just a suggestion).
> > > >
> > > > We appreciate your suggestion about including results from the Llama model. We will surely do our best to add these results to the final version of the paper, given the resources available to us.
> > > >
> > > > > Regarding Figure 5, I still expect to see conclusions about when Hybrid Similarity-Random is better and when "Hybrid Similarity-k-Means" is better. Instead of merely demonstrating "at least one of your methods outperforms the All Demonstrations setting".
> > > >
> > > > Thank you for your feedback. After analyzing our results in Tables 4 and 5, as well as considering the size of the pool of available demonstrations, **we would like to highlight that the hybrid similarity-random strategy is more effective than the hybrid similarity-k-means method when the pool of demonstrations is small and diversity is inherently low.** For example, in almost all experiments shown in Figure 5, which present results under a low-data regime, the hybrid similarity-random strategy outperforms the hybrid similarity-k-means method.
> > > >
> > > > However, **as the demonstration pool grows—and therefore becomes more diverse—the hybrid similarity-k-means method becomes more effective.** This can be seen in Figure 4 for the MetaTool, GSM Plus, and TREC datasets, where the pool sizes range from 5,000 to 20,000 demonstrations. In these cases, _the hybrid similarity-k-means method performs similarly to the hybrid similarity-random strategy._ **When the demonstration pool grows even further and diversity continues to increase,** such as in the ANLI dataset with 100,000 available demonstrations, **the hybrid similarity-k-means method can become more effective than the hybrid similarity-random strategy.**

---

### Official Review · Reviewer_wBuF · 2025-05-12

**Rating:** 6
**Confidence:** 3
**Ethics Flag:** 1

**Summary:**

The paper proposes a hybrid demonstration‑selection strategy for many‑shot in‑context learning that keeps most of the prompt cacheable while inserting a small block of query‑similar examples.

The manuscript is easy to follow and well‑structured. Experiments on four NLP tasks show promising accuracy–cost trade‑offs. Nevertheless, the significance of the findings is limited by evidence drawn from only two Gemini models and by inconsistent trends across datasets. Also, the evaluation lacks (a) variance reporting for the random‑selection baseline and (b) inclusion of a pure k‑means baseline, both of which curb statistical and methodological robustness.

**Minor comments**

- When the entire prompt is cached (i.e. random selection), the complexity of  attention calculation on the demonstration segment should reduce to O(1), yet Figure 2 depicts a different scaling; moreover, Fig. 2’s X‑axis metric and Fig. 4’s Y‑axis metric appear inconsistent.

**Reasons To Accept:**

Simple yet effective framework: Presents a computationally efficient many‑shot ICL method that leverages the LLM key–value cache; extending cached random (or k‑means‑selected) demonstrations with a small set of query‑similar examples is a natural and practical step, and the insights that can inspire further research.

**Reasons To Reject:**

- Limited generalizability of main claims
    - Experiments evaluate only two models from the same Gemini family, so the claimed compute–accuracy benefits may not hold for other LLM architectures or vendors.
    - The main claim appears to hold only for TREC and GSM‑Plus; MetaTool and ANLI exhibit a different trend, so generalization across tasks is not clear.
- Insufficient experiments
    - The random‑selection baseline reports no measure of variance (e.g., standard deviation across multiple random seeds), making it hard to judge statistical reliability.
    - A pure k‑means selection baseline is absent, so the incremental benefit of combining k‑means with similarity cannot be isolated.

---

> ### Author Response · Authors · 2025-06-02
> **Official Response by Authors (1/3)**
>
> Thank you for highlighting the effectiveness, practicality, and straightforward nature of our proposed demonstration selection strategies. We appreciate your thoughtful feedback and the opportunity to respond to your concerns in detail. Below, we address each of your concerns in the order they are raised.
>
> > When the entire prompt is cached (i.e. random selection), the complexity of  attention calculation on the demonstration segment should reduce to O(1), yet Figure 2 depicts a different scaling; moreover, Fig. 2’s X‑axis metric and Fig. 4’s Y‑axis metric appear inconsistent.
>
> Thank you for your input. We would like to clarify that **the complexity of attention computation, even when the prompt is cached, is not** $\mathcal{O}(1)$. Although the cached portion does not need to be recomputed, **each token from the downstream test sample still has to attend to all cached tokens as well as to itself.**
>
> More formally, when considering proportional inference cost, the complexity is given by $\mathcal{O}(t^2 + tc)$, where $t$ represents the number of tokens in the downstream test sample, and $c$ represents the number of cached tokens. Since $t$ is fixed across different $n$-shot ICL settings, the dominant term becomes $\mathcal{O}(tc)$. As $c$ increases, the dominant term simplifies to $\mathcal{O}(c)$. Therefore, complexity grows linearly with the number of cached tokens.
>
> *For a more detailed explanation, we would like to refer you to lines 207–215 of the paper, where this analysis is formally detailed.*
>
> > Experiments evaluate only two models from the same Gemini family, so the claimed compute–accuracy benefits may not hold for other LLM architectures or vendors.
>
> Thank you for your input. First, we would like to clarify that our experiments involve **two LLMs** and **seven different datasets,** tested under **two main scenarios:** a data-rich setting and a low-data regime. **This results in a total of 14 different experimental settings.**
>
> Additionally, we would like to highlight that **our choice of models and datasets was guided by the computational resources available to us, a strategy followed by all prominent prior works on many-shot ICL, such as Agarwal et al. (2024) and Bertsch et al. (2025).** For example, Agarwal et al. (2024) primarily used a single LLM (Gemini 1.5 Pro) in their experiments, while Bertsch et al. (2025) mostly focused on classification datasets. *Informed by these precedents, we did our best to carefully balance the number of models and datasets, ensuring diversity in both model size and task type. Specifically, we selected two LLMs of significantly different sizes, and our dataset choices span a range of tasks, from complex classification to tool use, reflecting practical applications valued by the community.*
>
> As a result, given our available resources, **we strongly believe that our study includes a sufficiently diverse set of models, datasets, and scenarios to effectively demonstrate the benefits of our proposed demonstration strategies in many-shot settings.**
>
> **Regarding the generalizability of our findings,** we would like to examine the *two core components* that underpin our proposed strategies: *(1) scaling the number of demonstrations, and (2) key-value (KV) caching.*
>
> For the first component—scaling demonstrations—we would like to refer to Bertsch et al. (2025), which showed that **ICL performance consistently improves as the number of demonstrations increases, across various LLMs, such as LLaMA models.** Since our method is also based on the same principle of increasing demonstrations, **our results also generalize similarly.** The second component, **KV caching, is inherently model-agnostic,** meaning the computational savings it provides are not tied to any specific LLM architecture. **Therefore, as these _two generalizable components_ form the foundation of our strategies, we are confident that our strategies generalize well to other LLM architectures as well.**

---

> > ### Author Response · Authors · 2025-06-02
> > **Official Response by Authors (2/3)**
> >
> > > The main claim appears to hold only for TREC and GSM‑Plus; MetaTool and ANLI exhibit a different trend, so generalization across tasks is not clear.
> >
> > Thank you for your input. To address your concern, we assume that by "main claim" in our paper, you are referring to the statement: **_"in many-shot settings, the influence of selection criteria (e.g., similarity) on performance diminishes as the number of demonstrations increases substantially, and beyond a certain point, their impact becomes nearly equivalent to that of random demonstrations (lines 46–49)."_**  If that is the case, then according to our results in Figure 4, the trend is actually the opposite of what you suggested: *while MetaTool and ANLI clearly follow our claim, it might seem that TREC and GSM-Plus behave differently at first glance.*
> >
> > If this is what you actually mean, we would like to clarify that our claim—that *“in many-shot settings, the influence of selection criteria (e.g., similarity) on performance diminishes as the number of demonstrations increases substantially”*—**always** includes the important qualifier: **“beyond a certain point.”** This qualifier is explicitly included in the relevant parts of the paper, specifically in lines 45–49 and 84–87, to emphasize that **performance approximation with the random selection strategy occurs only after a certain threshold.** Our results, shown in Figure 4, clearly support this claim, as do previous studies such as Agarwal et al. (2024) and Bertsch et al. (2025).
> >
> > **Regarding the TREC and GSM‑Plus datasets,** it can be seen in Figure 4 that the performance of the random selection strategy begins to converge to the similarity-based strategy as the number of demonstrations increases—particularly from 150 to 200. However, **for these datasets, the point at which similarity-based selection starts to closely approximate random selection seems to occur beyond 200 demonstrations. This convergence point is not visible in our current results due to the experimental limit of 200 demonstrations. Nevertheless, this does not invalidate our claim.** For example, *as shown by Bertsch et al. (2025) in Figure 2b, for the TREC dataset, further increasing the number of demonstrations leads to a close performance approximation between the random selection strategy and the similarity-based strategy.*
> >
> > **Therefore, our main claim in the paper is fully consistent with the reported results, and generalization across different tasks and models holds.**
> >
> > > The random‑selection baseline reports no measure of variance (e.g., standard deviation across multiple random seeds), making it hard to judge statistical reliability.
> >
> > Thank you for your input. We would like to note that *there are two widely accepted ways to ensure reliable experimental results.* One approach is to use a **non-zero temperature** during decoding, **run multiple trials to account for the probabilistic behavior of the model,** and **report the average performance along with the variance**—a method used by Agarwal et al. (2024). The other method is to use **greedy decoding (temperature set to zero) and fix the random seed across all experiments** to promote deterministic results—a method employed by Bertsch et al. (2025). Due to the high computational cost of many-shot ICL experiments and the wide range of $n$-shot ICL settings in our study, **we opted for the second method.**
> >
> > *Additionally, we would like to reiterate that we used a fixed random seed across all our experiments.* This ensures that the set of demonstrations selected for both the random selection strategy and the corresponding hybrid similarity-random strategy is identical, except for a few demonstrations that are replaced by the most similar ones to the downstream test sample. For example, if randomly selected demonstrations for 10-shot ICL are [Demo1, Demo2, Demo3, Demo4, Demo5, Demo6, Demo7, Demo8, Demo9, Demo10], then in *the corresponding hybrid similarity-random strategy, the same order of random demonstrations is preserved, but a few are replaced with the most similar demonstrations.* If we choose to include three of the most similar demonstrations to each test sample—Sim1, Sim2, and Sim3—the final demonstrations become [Sim1, Sim2, Sim3, Demo4, Demo5, Demo6, Demo7, Demo8, Demo9, Demo10]. **This design controls the stochasticity inherent in methods involving random selection, allowing a controlled, fair comparison when evaluating the effectiveness of our proposed strategies. Moreover, repeating this setup across different $n$-shot ICL settings further addresses concerns related to randomness in stochastic methods.**
> >
> > **Given these considerations and taking these design details into account, we firmly believe that our experimental design provides a reliable and fair comparison between our proposed methods and the baselines.**

---

> > ### Author Response · Authors · 2025-06-02
> > **Official Response by Authors (3/3)**
> >
> > > A pure k‑means selection baseline is absent, so the incremental benefit of combining k‑means with similarity cannot be isolated.
> >
> > Thank you for your feedback. First, we would like to emphasize that **our study already includes two widely adopted baselines used in many-shot ICL settings, as employed in the previous studies by Agarwal et al. (2024) and Bertsch et al. (2025).**
> >
> > Second, we would like to emphasize that using **$k$-means as a baseline in the many-shot ICL setting is not a common practice,** and to the best of our knowledge, **it has not been used in prominent prior studies, such as Agarwal et al. (2024) and Bertsch et al. (2025).** Furthermore, as we pointed out in several parts of our paper (e.g., lines 31–32 and 82–87), and as demonstrated in aforementioned previous works, **the effectiveness of selection criteria such as similarity or diversity (as employed in $k$-means) diminishes as the number of demonstrations increases.** *Beyond a certain point,* demonstrations selected using a specific criterion tend to resemble those selected at random. This can be seen in our results in Figure 4, where the performance of similarity-based and random selection strategies becomes nearly identical at higher demonstration counts. **The same pattern holds for diversity-based methods such as $k$-means: _as the number of demonstrations grows, the selected demonstrations increasingly resemble those chosen randomly._ Therefore, from this perspective, a $k$-means baseline is unlikely to offer insights beyond those already provided by the similarity-based method.**
> >
> > *Nevertheless, to fully address this concern, we will include the results from the $k$-means baseline in the final version of the paper.*
> >
> > ---
> >
> > Lastly, we hope our explanations have addressed your concerns effectively. While we have aimed to cover every aspect thoroughly, we are happy to provide further clarification or expand on any topics if you still have questions or feel our responses were not convincing.
> >
> > ---
> >
> > **References:**
> >
> > [1] Many-Shot In-Context Learning (Agarwal et al., NeurIPS 2024)
> >
> > [2] In-Context Learning with Long-Context Models: An In-Depth Exploration (Bertsch et al., NAACL 2025)

---

> > > ### Author Response · Authors · 2025-06-10
> > > **Kind Reminder about Our Rebuttals**
> > >
> > > Dear Reviewer wBuF,
> > >
> > > We would like to thank you for taking the time to review our paper. As the discussion period is coming to an end, we would like to kindly remind you that we have _carefully_ and _thoroughly_ responded to all your concerns. We value your feedback and would appreciate it if you could provide any additional input or confirm whether our responses have addressed your concerns.
> > >
> > > Thank you for your time and attention.
> > >
> > > Best regards,
> > > The Authors

---

> > > > ### Comment · Reviewer_wBuF · 2025-06-10
> > > > **Thank the reviewer for the clarification.**
> > > >
> > > > Dear Authors,
> > > >
> > > > I truly appreciate the author's feedback on my review comments. My concerns have been addressed. I hope the authors clarify these points in the final version, as the reader may experience similar confusion.

---

> > > > > ### Author Response · Authors · 2025-06-10
> > > > > **Official Response by Authors**
> > > > >
> > > > > We would like to sincerely thank you for your participation in the discussion period. We are pleased to hear that our responses have addressed your concerns. Furthermore, we would like to assure you that we will carefully incorporate all of your points and feedback into the final version of the paper.

---

### Official Review · Reviewer_MXr4 · 2025-05-14

**Rating:** 7
**Confidence:** 3
**Ethics Flag:** 1

**Summary:**

This study proposed a new method for few shot exemplar selection under in-context learning. The method contains two sub-approaches - the first one compare the similarity between the exemplars to the ones in the test set utilizing embeddings; the second one applying centroids derived from test sample representations via k-means clustering for exemplar selection.

**Questions To Authors:**

Please see the above section for questions and comments.

**Reasons To Accept:**

Developing new methods for exemplar selection is an important topic for prompt engineering. Effective approaches could potentially benefit downstream applications. The study proposed a straightforward approach for selecting prompts. Results on two LLMs also show performance improvement compared to the baseline methods for exemplar selection.

**Reasons To Reject:**

1. The study evaluated the methods on two models - Gemini Pro and Flash to further test the generalizability of the method. It would be helpful if more open-source LLMs could be tested on the datasets.

2. Besides the testing of the current 4 datasets, it might be helpful if more datasets, especially the domain-specific ones, could be included for further experiments.

---

> ### Author Response · Authors · 2025-06-02
> **Official Response by Authors (1/1)**
>
> Thank you for pointing out the effectiveness, practicality, and straightforward nature of our proposed strategies. We are grateful for your constructive input. We appreciate the opportunity to address your concerns. To do so, we have provided our responses in the same order in which your concerns are listed.
>
> > 1. The study evaluated the methods on two models - Gemini Pro and Flash to further test the generalizability of the method. It would be helpful if more open-source LLMs could be tested on the datasets.
>
> Thank you for your input. First, we would like to clarify that our experiments involve **two LLMs** and **seven different datasets,** tested under **two main scenarios:** a data-rich setting and a low-data regime. **This results in a total of 14 different experimental settings.**
>
> Additionally, we would like to highlight that **our choice of models and datasets was guided by the computational resources available to us, a strategy followed by all prominent prior works on many-shot ICL, such as Agarwal et al. (2024) and Bertsch et al. (2025).** For example, Agarwal et al. (2024) primarily used a single LLM (Gemini 1.5 Pro) in their experiments, while Bertsch et al. (2025) mostly focused on classification datasets. *Informed by these precedents, we did our best to carefully balance the number of models and datasets, ensuring diversity in both model size and task type. Specifically, we selected two LLMs of significantly different sizes, and our dataset choices span a range of tasks, from complex classification to tool use, reflecting practical applications valued by the community.*
>
> As a result, given our available resources, **we strongly believe that our study includes a sufficiently diverse set of models, datasets, and scenarios to effectively demonstrate the benefits of our proposed demonstration strategies in many-shot settings.**
>
> **Regarding the generalizability of our findings,** we would like to examine the *two core components* that underpin our proposed strategies: *(1) scaling the number of demonstrations, and (2) key-value (KV) caching.*
>
> For the first component—scaling demonstrations—we would like to refer to Bertsch et al. (2025), which showed that **ICL performance consistently improves as the number of demonstrations increases, across various LLMs, including open-weight LLaMA models.** Since our method is also based on the same principle of increasing demonstrations, **our results also generalize similarly.** The second component, **KV caching, is inherently model-agnostic,** meaning the computational savings it provides are not tied to any specific LLM architecture. **Therefore, as these _two generalizable components_ form the foundation of our strategies, we are confident that our strategies generalize well to other LLM architectures as well.**
>
> > 2. Besides the testing of the current 4 datasets, it might be helpful if more datasets, especially the domain-specific ones, could be included for further experiments.
>
> Thank you for your input. We addressed this before. Please find our response above.
>
> ---
>
> Lastly, we hope our responses have effectively addressed your concerns. If you have any further questions or need additional clarification, please let us know. We would be more than happy to discuss any aspect in greater detail.
>
> ---
>
> **References:**
>
> [1] Many-Shot In-Context Learning (Agarwal et al., NeurIPS 2024)
>
> [2] In-Context Learning with Long-Context Models: An In-Depth Exploration (Bertsch et al., NAACL 2025)

---

### Official Review · Reviewer_KMMK · 2025-05-17

**Rating:** 6
**Confidence:** 3
**Ethics Flag:** 1

**Summary:**

The authors address the high inference cost of many-shot in-context learning (ICL) when long-context LLMs ingest hundreds of demonstrations. The key idea is to keep most of the prompt cacheable while still tailoring a small slice to each test instance:
- Hybrid Similarity + Random: For an n-shot prompt, pick s (e.g., 20) demonstrations most similar to the test query and prepend them to r = n – s randomly chosen demos that remain fixed across all queries and can therefore be cached.
- Hybrid Similarity + k-Means: Replace the random cache with k demonstrations chosen via k-means over test-query embeddings, aiming for higher diversity.

Experiments on Gemini Pro and Gemini Flash across four datasets (ANLI-R3, TREC-50, GSM Plus, MetaTool) show that both hybrids consistently match—or surpass—full per-query similarity selection while cutting the quadratic attention cost to near-linear (thanks to caching). Additional tests on three BBH tasks demonstrate benefits in low-data regimes.

**Reasons To Accept:**

- Practicality & Simplicity – The strategies are easy to implement and integrate smoothly with existing key–value caching.
- Clear Cost Analysis – The paper formalises proportional cost and backs it with empirical timing.

**Reasons To Reject:**

- Limited Model and Task Generalization: The evaluation is restricted to two models from the Gemini family and four datasets. This narrow scope raises concerns about the generalizability of the findings to other LLM architectures (e.g., LLaMA) and task types, particularly domain-specific or diverse NLP benchmarks.
- Incomplete and Inconclusive Experimental Design: Key baselines are missing (e.g., pure k-means), and no variance is reported for stochastic methods such as random selection. These omissions hinder a reliable assessment of the method’s effectiveness and statistical significance.
- Inconsistent Claims and Observations: Some claims in the paper are contradicted by the results—for example, the assertion that similarity-based selection approximates random selection conflicts with large accuracy differences seen in certain datasets (e.g., TREC). This undermines the internal coherence of the study.

---

> ### Author Response · Authors · 2025-06-02
> **Official Response by Authors (1/3)**
>
> Thank you for recognizing the practicality and simplicity of our method, and we are happy you found our cost analysis clear. We also appreciate your thoughtful and detailed feedback. We are grateful for the opportunity to address your concerns. Below, our responses are provided in the same order as your comments.
>
> > Limited Model and Task Generalization: The evaluation is restricted to two models from the Gemini family and four datasets. This narrow scope raises concerns about the generalizability of the findings to other LLM architectures (e.g., LLaMA) and task types, particularly domain-specific or diverse NLP benchmarks.
>
> Thank you for your input. First, we would like to clarify that our experiments involve **two LLMs** and **seven different datasets,** tested under **two main scenarios:** a data-rich setting and a low-data regime. **This results in a total of 14 different experimental settings.**
>
> Additionally, we would like to highlight that **our choice of models and datasets was guided by the computational resources available to us, a strategy followed by all prominent prior works on many-shot ICL, such as Agarwal et al. (2024) and Bertsch et al. (2025).** For example, Agarwal et al. (2024) primarily used a single LLM (Gemini 1.5 Pro) in their experiments, while Bertsch et al. (2025) mostly focused on classification datasets. *Informed by these precedents, we did our best to carefully balance the number of models and datasets, ensuring diversity in both model size and task type. Specifically, we selected two LLMs of significantly different sizes, and our dataset choices span a range of tasks, from complex classification to tool use, reflecting practical applications valued by the community.*
>
> As a result, given our available resources, **we strongly believe that our study includes a sufficiently diverse set of models, datasets, and scenarios to effectively demonstrate the benefits of our proposed demonstration strategies in many-shot settings.**
>
> **Regarding the generalizability of our findings,** we would like to examine the *two core components* that underpin our proposed strategies: *(1) scaling the number of demonstrations, and (2) key-value (KV) caching.*
>
> For the first component—scaling demonstrations—we would like to refer to Bertsch et al. (2025), which showed that **ICL performance consistently improves as the number of demonstrations increases, across various LLMs, including LLaMA models.** Since our method is also based on the same principle of increasing demonstrations, **our results also generalize similarly.** The second component, **KV caching, is inherently model-agnostic,** meaning the computational savings it provides are not tied to any specific LLM architecture. **Therefore, as these _two generalizable components_ form the foundation of our strategies, we are confident that our strategies generalize well to other LLM architectures as well.**

---

> > ### Author Response · Authors · 2025-06-02
> > **Official Response by Authors (2/3)**
> >
> > > Incomplete and Inconclusive Experimental Design: Key baselines are missing (e.g., pure k-means), and no variance is reported for stochastic methods such as random selection. These omissions hinder a reliable assessment of the method’s effectiveness and statistical significance.
> >
> > Thank you for your feedback. First, we would like to emphasize that **our study already includes two widely adopted baselines used in many-shot ICL settings, as employed in the previous studies by Agarwal et al. (2024) and Bertsch et al. (2025).**
> >
> > Second, we would like to emphasize that using **$k$-means as a baseline in the many-shot ICL setting is not a common practice,** and to the best of our knowledge, **it has not been used in prominent prior studies, such as Agarwal et al. (2024) and Bertsch et al. (2025).** Furthermore, as we pointed out in several parts of our paper (e.g., lines 31–32 and 82–87), and as demonstrated in aforementioned previous works, **the effectiveness of selection criteria such as similarity or diversity (as employed in $k$-means) diminishes as the number of demonstrations increases.** *Beyond a certain point,* demonstrations selected using a specific criterion tend to resemble those selected at random. This can be seen in our results in Figure 4, where the performance of similarity-based and random selection strategies becomes nearly identical at higher demonstration counts. **The same pattern holds for diversity-based methods such as $k$-means: _as the number of demonstrations grows, the selected demonstrations increasingly resemble those chosen randomly._ Therefore, from this perspective, a $k$-means baseline is unlikely to offer insights beyond those already provided by the similarity-based method.** *Nevertheless, to fully address this concern, we will include the results from the $k$-means baseline in the final version of the paper.*
> >
> > **Regarding the absence of variance reporting for random selection,** we would like to note that *there are two widely accepted ways to ensure reliable experimental results.* One approach is to use a **non-zero temperature** during decoding, **run multiple trials to account for the probabilistic behavior of the model,** and **report the average performance along with the variance**—a method used by Agarwal et al. (2024). The other method is to use **greedy decoding (temperature set to zero) and fix the random seed across all experiments** to promote deterministic results—a method employed by Bertsch et al. (2025). Due to the high computational cost of many-shot ICL experiments and the wide range of $n$-shot ICL settings in our study, **we opted for the second method.**
> >
> > *Additionally, we would like to reiterate that we used a fixed random seed across all our experiments.* This ensures that the set of demonstrations selected for both the random selection strategy and the corresponding hybrid similarity-random strategy is identical, except for a few demonstrations that are replaced by the most similar ones to the downstream test sample. For example, if randomly selected demonstrations for 10-shot ICL are [Demo1, Demo2, Demo3, Demo4, Demo5, Demo6, Demo7, Demo8, Demo9, Demo10], then in *the corresponding hybrid similarity-random strategy, the same order of random demonstrations is preserved, but a few are replaced with the most similar demonstrations.* If we choose to include three of the most similar demonstrations to each test sample—Sim1, Sim2, and Sim3—the final demonstrations become [Sim1, Sim2, Sim3, Demo4, Demo5, Demo6, Demo7, Demo8, Demo9, Demo10]. **This design controls the stochasticity inherent in methods involving random selection, allowing a controlled, fair comparison when evaluating the effectiveness of our proposed strategies. Moreover, repeating this setup across different $n$-shot ICL settings further addresses concerns related to randomness in stochastic methods.**
> >
> > **Given these considerations and taking these design details into account, we firmly believe that our experimental design provides a reliable and fair comparison between our proposed methods and the baselines.**

---

> > ### Author Response · Authors · 2025-06-02
> > **Official Response by Authors (3/3)**
> >
> > > Inconsistent Claims and Observations: Some claims in the paper are contradicted by the results—for example, the assertion that similarity-based selection approximates random selection conflicts with large accuracy differences seen in certain datasets (e.g., TREC). This undermines the internal coherence of the study.
> >
> > Thank you for your input. We would like to clarify that our claim—that *“similarity-based selection approximates random selection”*—is **_always_** accompanied by the important qualifier **_“beyond a certain point.”_** This qualifier is explicitly included in the relevant parts of the paper, specifically in lines 45–49 and 84–87, to emphasize that **this approximation occurs only after a certain threshold.** Our results, shown in Figure 4, clearly support this claim, as do previous studies such as Agarwal et al. (2024) and Bertsch et al. (2025).
> >
> > **Regarding the TREC dataset,** it can be seen in Figure 4 that the performance of the random selection strategy begins to converge to the similarity-based strategy as the number of demonstrations increases—particularly from 150 to 200. However, **for this dataset, the point at which similarity-based selection starts to closely approximate random selection seems to occur beyond 200 demonstrations. This convergence point is not visible in our current results due to the experimental limit of 200 demonstrations. Nevertheless, this does not invalidate our claim,** and the TREC dataset still falls within the scope of our assertion. In fact, *as shown in the work of Bertsch et al. (2025), specifically in Figure 2b, further increasing the number of demonstrations leads to a close approximation between the two strategies for this dataset.* **Therefore, our claims in the paper are fully consistent with the reported results, and there is no contradiction between our assertion and the data.**
> >
> > ---
> >
> > Finally, we hope our responses have effectively addressed your concerns. We would also be happy to discuss any points in further detail if you remain unconvinced by our answers.
> >
> > ---
> >
> > **References:**
> >
> > [1] Many-Shot In-Context Learning (Agarwal et al., NeurIPS 2024)
> >
> > [2] In-Context Learning with Long-Context Models: An In-Depth Exploration (Bertsch et al., NAACL 2025)

---

> > ### Comment · Reviewer_MXr4 · 2025-06-09
> >
> > Thank you for the clarification and response. I have raised my score.

---

> > > ### Author Response · Authors · 2025-06-10
> > > **Official Response by Authors**
> > >
> > > We would like to sincerely thank you for your participation during the discussion period. We are pleased that our responses have addressed your concerns.

---

> ### Author Response · Authors · 2025-06-10
> **Kind Reminder about Our Rebuttals**
>
> Dear Reviewer KMMK,
>
> We would like to thank you for taking the time to review our paper. As the discussion period is coming to an end, we would like to kindly remind you that we have _carefully_ and _thoroughly_ responded to all your concerns. We value your feedback and would appreciate it if you could provide any additional input or confirm whether our responses have addressed your concerns.
>
> Thank you for your time and attention.
>
> Best regards,
> The Authors

---

### Decision · Program_Chairs · 2025-07-08

**Decision:**

Accept

**Comment:**

The paper provides solid empirical grounding for a selection of clustering-based methods for choosing ICL examples. This is an important, impactful, and poorly understood area. While there are limited theoretical contributions, these practical results provide a good basis for further analysis.

The authors provide extensive responses in discussion that are generally acceptable to reviewers, who show an admirable willingness to be persuaded. These set a solid agenda for a revised final version.